# Development and Validation of a Brief Scale of Vengeful Tendencies (BSVT-11) in a Mexican Sample

**DOI:** 10.3390/bs12070215

**Published:** 2022-06-28

**Authors:** Ana Lorena Flores-Camacho, Diana Laura Castillo-Verdejo, Julio C. Penagos-Corzo

**Affiliations:** Department of Psychology, Universidad de las Américas Puebla, San Andrés Cholula 72810, Mexico; ana.floresco@udlap.mx (A.L.F.-C.); diana.castillovo@udlap.mx (D.L.C.-V.)

**Keywords:** revenge, resentment, vindictive cognition, factor analysis, retaliation

## Abstract

The development and analysis of psychometric properties of a brief scale that assesses vengeful tendencies (BSVT-11) is presented. A three-dimensional model is proposed: (1) resentment, (2) planning of revenge, and (3) justification of revenge. Two studies were conducted for this purpose: one was carried out with a sample of 478 participants, to evaluate the content validity, factorial structure, reliability, and invariance of the instrument; the other was conducted with a sample of 308 participants, to determine the concurrent validity of the BSVT-11. The data indicated adequate reliability (ω = 0.877), optimal fit (CFI = 1.0, TLI = 1.0) according to the dimensions proposed, and invariance between genders (*p* = 0.893). Concurrent validity data yield significant correlation values (*p* < 0.001) ranging from 0.522 to 0.804 in the analyses between the BSVT and other scales. The results allow us to present a brief instrument with good psychometric properties with possibilities for use in basic and translational science.

## 1. Introduction

Revenge is an important global phenomenon [1] in interpersonal relationships [2] that can be described as the attempt to impose suffering on those who have done harm [3]. More recent approaches identify it as a personal response to unfair treatment [4], or as harmful acts performed in return for a perceived wrong [5]. From this point of view, revenge has been defined as the attempt to repair an interpersonal harm by voluntarily committing an aggressive action against the offender [6]. Returning the harm received provides social approval for revenge, and at the same time, discourages maltreatment [7].

Vindictive acts can produce significant personal risks through subsequent retaliation [5], which can cause people to keep thinking about the transgressor whom they have punished, thus prolonging their hedonic reactions to the transgression [8]. However, despite the high cost involved, people consider it worthwhile to take revenge [1]. Moreover, for those who seek revenge, it is a moral imperative and an act of justice [9]. Therefore, if someone considers it possible, he or she will continue to have ruminative thoughts that will be part of the motivation for retaliation [10]. In fact, rumination plays an important role in revenge, making forgiveness difficult [11]. Similarly, memories related to anger also contribute to thoughts of revenge [12].

When personal aggression occurs, an irresistible desire for revenge may follow [13]. This compelling desire for revenge includes seeking redress for the harm, and satisfaction in seeking forms of revenge [14]. Although acts of revenge may be considered impulsive [3], it has been noted that revenge may take place over an extended period of time [15]. After a transgression, some people experience persistent negative motivation for days, weeks, months [16], or even years [17]. Such motivation, in addition to being fueled by the rumination of revenge, has emotional correlates [18]. In this sense, revenge involves components of planning, in addition to emotional satisfaction components, which vary over time [2]. In fact, it has been shown that revenge behavior is related to the activity of the dorsolateral prefrontal cortex (dlPFC) [19], which plays a significant role in planning tasks [20].

Suffering unfair treatment or mistreatment can generate resentment that leads to revenge [21]. In fact, in violent actions such as terrorism, a relationship can be found between perceived humiliation, resentment, and revenge on the part of the perpetrator [22]. This implies that resentment becomes a motivator for revenge [23]. In this sense, aggressive and hostile reactions develop in response to unjust events, which are then likely to lead to resentment and the desire for revenge [24]. Further, mismanaged resentment will lead to revenge, not aggressive behavior alone [25]. Resentment is anger based on the assumption of having suffered some form of mistreatment, and invokes a moral concept (a recognized right or wrong) as an explanation for the feeling one has experienced [26].

In addition to resentment and thoughts of revenge, it is also important to consider the victim’s moral justifications of his or her emotions and behaviors related to his or her desire for revenge. In this sense, it has been pointed out that the acts perceived as severe [27] and/or aggressive [28], according to the moral values that a subject has, are those that define the intensity or presentation of revenge [1]. Generally, acts of revenge are only considered appropriate when there is an acceptable justification [5].

The goals and/or purpose of revenge can be multiple [5]. However, it has been argued that acts of revenge have the purpose of teaching the transgressor a moral lesson [6]. At the same time, some kind of reward is sought by giving the aggressor what is believed to be deserved [4]. In this way, the victim feels satisfied with seeing the transgressor suffer [4], and feels as though they have gotten even or have balanced the scales [6]. Though various emotions are associated with vindictive activity [5], the literature has indicated that people report feeling much better after the opportunity to punish [8]. Similarly, it has been claimed that engaging in vengeful actions has beneficial effects for the avenger [4]. Although it is possible to experience positive feelings after revenge, these are mild and are tempered by emotional experiences of guilt, remorse, anger, and fear [5]. This is because punishers continue to ruminate about their aggressor [8].

There are several instruments used to assess vengeance. These include relatively extensive scales, such as the Vengeance Scale [29] or the Multidimensional Revenge Attitudes Inventory-21 [14], in addition to scales with a broader meaning, such as the Transgression-Related Interpersonal Motivations Inventory [30,31] or the Anger Rumination Scale (ARS) [32]. However, although such instruments report adequate psychometric properties, their values in confirmatory factor analyses frequently do not exceed the optimal cut-off points indicated by Schermelleh-Engel et al. [33]. Even lower, usually accepted values, such as those indicated by Barret [34], or by Hu and Bentler [35], are not reached for all indicators. This suggests that the models require improvement. In addition, there is a lack of adaptations for populations such as the one studied here.

In relation to the population on which this study is focused, to date, it was only possible to find instruments that indirectly address revenge. For example, in the Anger Thinking Scale [36,37], one of the factors is about revenge. However, the test is not specifically designed for the construct of revenge, and, therefore, the measurement is unidimensional (e.g., thoughts of revenge), and is in relation to anger. This is also the case for the Displaced Aggression Questionnaire, which has been validated for Spanish-speaking countries, likely with a Spanish sample [38]. Although one of its factors is about revenge, its assessment is in the context of another construct and not of revenge itself. The same is applicable to the Anger Rumination Scale, which has also been adapted for Spanish speakers [32]. Although the Anger Rumination Scale measures revenge, it is also a unidimensional measurement (e.g., thoughts of revenge) and does so in relation to another construct: anger rumination. In addition, the model presented here focuses on three dimensions that maintain revenge: resentment, planning revenge, and justification of revenge. In this sense, there are no instruments that assess revenge under the proposed model. Therefore, the aim of the present study is to analyze the evidence of validity and reliability of a brief scale of revenge tendencies. Specifically, the aim is to find three factors: resentment, justification of revenge, and planning of revenge.

## 2. Study 1. Factorial Structure and Main Psychometric Properties

Study 1 analyzes the psychometric properties of the BSVT-11, evaluating the content validity ratio, the factorial structure through exploratory factor analysis, the structural validity through confirmatory factor analysis, the reliability both in terms of internal consistency and temporal stability, and the invariance of the instrument.

### 2.1. Materials and Methods

#### 2.1.1. Participants

Based on an invitation posted on social networks, participants who responded positively to the invitation were selected (*n* = 555). Of this sample, 77 participants were discarded because they were under 18 years of age or for having randomly answered the questionnaire. This may include having responded affirmatively to items provided for random response control (e.g., “sometimes I do not remember my name”) or for excessively short response time when answering the instrument that implied not having read the items. The final sample consisted of 478 participants—334 of whom were female, and 144 of whom were males. All participants were Mexicans from different urban settings of Mexico. Eighty-two percent came from eight central states, but mainly from Mexico City, the State of Mexico, and Puebla. Fifteen percent came from seven southern and southeastern states, mainly from the state of Veracruz. The remaining 4% came from eight states in northern Mexico. The mean age was 28.02 years (SD = 13.11). The age range of participants spanned between 18 years to 82 years. The sample was divided into two subsamples: one (SS1) for the exploratory factor analysis, and the other (SS2) for the confirmatory factor analysis. The SS1 sample consisted of 183 women and 67 men. The mean age of the females was 27.6 (SD = 12.6), and the mean age of the males was 27.3 (SD = 12.9). On the other hand, SS2 consisted of 151 females and 77 males. The mean age of the males was 30.3 (SD = 13.7), and the mean age of the females was 27.6 (SD = 13.5). All participants gave informed consent, and were treated in accordance with the Ethical Code of the Psychologist in Mexico [39] and with the Declaration of Helsinki [40].

#### 2.1.2. Instrument

*Brief Scale of Vengeful Tendencies* (BSVT): Instrument developed for this study, composed of 11 items created to assess three dimensions: resentment, revenge planning, and revenge justification. Participants were asked to choose one of five response options on a Likert scale, where the lowest value is “Strongly disagree” (1), and in ascending order: “Disagree” (2), “Indifferent” (3), “Agree” (4), “Strongly agree” (5). At the beginning of the scale, the participants read the following instruction: “*Below will be a series of statements, please indicate the option that best describes you*”. Participants were made aware in the informed consent that the test was intended to assess different attitudes related to retaliation or responses to unfair treatment.

#### 2.1.3. Procedure

The psychometric study began with the initial development of the items, which were evaluated by a group of experts, and subsequently tested in a pilot study. The resulting items were applied to the general sample through an online form.

This study validated the instrument with the three types of standard validity: content, construct, and criterion. In relation to these three types of validity, Cohen et al. [41] has indicated that content validity refers to how adequately a test is a sample of behavior representative of the universe of behaviors for which the test was designed. Construct validity refers to the judgment of how adequately a test score can be used to infer a person’s most likely position on a measure of interest. Finally, criterion validity is defined as the judgment about the appropriateness of the inferences made from the scores obtained in the test with respect to the variable called construct. For the first case, the content validity ratio, described in detail below, was used. For the second case, concurrent validity was used, which was addressed in the second study. Finally, construct validity was addressed first by structural analysis of the construct, then by exploratory factor analysis, and finally by confirmatory factor analysis.


*Item development*


The items were generated based on the recommendations of Boateng et al. [42]. This involved using a logical or deductive partitioning method by means of reviewing the literature and evaluating indicators of the construct to be measured. This review made it possible to elaborate the items based on three dimensions: resentment [22,23], justification of revenge [1,5], and planning revenge [4,6]. Based on the above, 15 items were drafted, five per dimension, which were evaluated by experts to determine content validity.


*Content validity*


For content validity, five experts were invited to serve as judges. These experts were university professors, all of whom have had lines of research or subjects they taught that are related to the topic under study. The experts evaluated the relevance of each item using a three-level criterion [43]: (a) the item is essential, (b) the item is useful but not essential, and (c) the item is not relevant. The content validity ratio was determined based on the Tristan-Lopez algorithm [44], which resulted in the elimination of four items, leaving a total of 11 items.


*Statistical analysis*


Statistical analyses were performed in SPSS v 24 [45], AMOS v 24 [46], and Jamovi v 2.2.5. For the exploratory factor analysis, principal axis factoring with PROMAX rotation was used due to the consideration that the factors were correlated. For the factor extraction process, the main criteria were: eigenvalue of the parallel analysis greater than the simulated random values [47], eigenvalue greater than 1 [48], and factor loadings greater than 0.40. The confirmatory factor analysis was evaluated with the comparative fit index (CFI), the Tucker–Lewis index (TCI), and the root mean square error of approximation (RMSEA). Regarding the CFI and TLI indices, there is general agreement to use cut-off points of 0.95 as an indicator of optimal fit, as well as values below 0.06 for the RMSEA [34,35]. Internal consistency was assessed by McDonald’s omega, and temporal stability was determined by Pearson’s correlation. The omega coefficient was used because, unlike the alpha coefficient, omega is not affected by the number of items, it works with factor loadings, and it is considered superior to the alpha coefficient [49].

### 2.2. Results

#### 2.2.1. Exploratory Factor Analysis

To determine the factorial structure of the instrument, an exploratory factor analysis was performed. This technique is used to empirically confirm the conceptual structure previously established [50]. Prior to the exploratory factor analysis (EFA), a homogeneity analysis was performed to verify that all items had scores above 0.30 on the corrected total item correlation. This was confirmed with values ranging from 0.455 to 736. Once homogeneity was confirmed, Kaiser–Mehyer–Olkin (KMO) (KMO = 0.889) and Bartelett’s sphericity (χ^2^ = 1219, df = 55, *p* < 0.001) tests were performed, which indicated the relevance of continuing with the EFA. Parallel analysis, eigenvalue values, and factor loadings (Table 1) indicated three factors as proposed. Factor 1 (F1), with an eigenvalue of 5.141, explained 23.9% of the variance. Factor 2 (F2) obtained an eigenvalue of 1.067 and explained 16.5% of the variance. Factor 3 (F3) explained 13.1% of the variance and obtained an eigenvalue of 1.0. Thus, the three factors together explained 53.5% of the total variance. The inter-factor correlations were moderate. F1–F2 = 0.690, F1–F3 = 0.630, F2–F3 = 0.614. The minimum communality extracted was 0.312, and the maximum was 0.730.

#### 2.2.2. Confirmatory Factor Analysis

Confirmatory factor analysis allows the testing of a model built in advance [50]. Confirmatory factor analysis was performed on the SS2 subsample using the maximum likelihood estimation method (Figure 1). Model fit indices suggested adequate fit, χ^2^(41) = 37.3, *p* = 0.637 (χ^2^/DF = 0.910), with perfect levels, CFI = 1.0 TLI = 1.0 and RMSEA = 0.00 (90% confidence interval, 0.00 lower and 0.039 upper). The goodness-of-fit index, parsimony, and root mean square residual also showed acceptable levels (GFI = 0.971, AGFI = 0.954, PNFI = 0.719, PCFI = 0.745, SRMR = 0.026). Other values, reported less frequently in the literature, also show good fits: NFI = 0.964, RFI = 0.952, PGFI = 603, NFI = 0.964, PGFI = 603, NFI = 0.964, RFI = 0.952.

#### 2.2.3. Reliability Analysis

*Internal consistency*.

Internal consistency was evaluated using McDonald’s omega. The sample for this analysis included the total of samples SS1 and SS2 (N = 478) with the characteristics previously described in the participants section of this paper. Factor F1 obtained a ω = 0.863 and the average interitem correlation (AIC) was = 0.609; for F2, ω = 0.754 and AIC = 0.422; whereas for F3, ω = F3 = 0.684 and AIC = 0.408. Total BSVT obtained a ω = 0.877 and an AIC = 0.403. From the confirmatory factor analysis data, the composite reliability was obtained, which in all factors are > 0.70, which is the standard cut-off point for this measure: F1 = 0.782, F2 = 0.853, and F3 = 0.702.


*Temporal stability.*


From the total sample (SS1 and SS2), data were only available for 78 participants who could be located 30 days after the first application. With this sample, an analysis of the relationship between a pre- and a post-application was performed. The data indicated a significant moderate correlation (r = 0.871, *p* < 0.001).

#### 2.2.4. Descriptive Data and Analysis of Differences

Table 2 shows the arithmetic means (M) and standard deviations (SD) for women and men, as well as for the sample in general. Because descriptively higher values were observed for the male population, it was decided to compare the results for men and women. Differences were found in all factors: F1 (t(476) = 5.10, *p* < 0.001, d = 0.508, 95% CI (0.309 0.707)), F2 (t(476) = 3.99, *p* < 0.001, d = 3.98, 95% CI (0. 20 0.596)), F3 (t(476) = 4.25, *p* < 0.001, d = 0.423, 95% CI (0.225 0.621)), and also in total BSVT (t(476) = 5.25, *p* < 0.001, d = 0.524, 95% CI (0.324 0.723)).

#### 2.2.5. Analysis of Invariance

The structure of the BSVT was analyzed between males and females (Model 1, unconstrained (M1UC)). The data indicated adequate fit (CFI = 0.990, GFI = 0.934, TLI = 0.987, RMSEA = 0.023). Subsequently, this analysis was conducted on a fully constrained model (M2FC). With this model, the fit indices were also adequate (CFI = 0.996, GFI = 0.930, TLI = 0.995, RMSEA = 0.014). The differences in chi-square between the two models indicated invariance (X2_M1UC_ = 91.5, df = 82, X2_M2FC_ = 97.2, df = 93, *p* = 0.893).

### 2.3. Discussion

The data allow us to confirm that the proposed scale has an optimal fit, congruent with the proposed model. Three salient factors were found in the EFA, and were confirmed in the CFA. These factors correspond to the three dimensions proposed: F1 corresponds to planning revenge, F2 corresponds to resentment, and F3 corresponds to justification of revenge. Reliability data also show satisfactory results, with adequate internal consistency and temporal stability.

The correlations between F1 (planning) and F2 (resentment) were the highest among the individual BSVT factors. Resentment is both moral and affective [26]. It is likely that revenge planning requires an emotional force to drive it, and that force may be resentment. In fact, resentment converts more directly to revenge in people with higher rates of violence than less violent people [25].

It has been pointed out that people who take revenge or retaliate simultaneously have cognitive dissonance derived from their actions [51]. For this reason, the correlations of F1 and F3 (justification) are relevant, as they would suggest that people require cognitive efforts to justify their revenge intentions.

The findings that men were more vengeful than women are congruent with other results [52,53]. Though not all studies are consistent that men are more vengeful than women, there is strong support that men report more vindictiveness than women [52]. It is important to note that despite such differences, the inventory showed invariance with respect to gender.

## 3. Study 2 Concurrent Validity

To assess the concurrent validity of the BSVT, it was proposed that there would be relationships between three scales conceptually linked to the dimensions assessed by the BSVT-11. The following hypotheses were, therefore, posed: (1) there will be a positive relationship between the revenge planning dimension of the BSVT-11 and the revenge motivation factor of the Transgression-Related Interpersonal Motivations (TRIM-12) [30]; (2) there will be a positive relationship between the resentment dimension of the BSVT-11 and the interpersonal resentment factor of the Collectivist-Sensitive Trait Forgivingness Scale (TFS-CS) [54]; (3) the revenge justification dimension would have a positive relationship with the self-justice factor of the Potential Predatory Violence Inventory (PPVI) [55]; (4) positive relationships were expected between the total BSVT-11 and each of the concurrent validity scales used.

### 3.1. Materials and Methods

#### 3.1.1. Participants

The sample consisted of 308 adults who agreed to participate in the research based on an invitation posted on the university’s social networks. The initial sample was composed of 310 participants, but two participants were discarded because the time they used to answer the scales was much less than the time it would have taken to read the questionnaire. Two-hundred and thirty-three of the participants were female, and 75 were male. The mean age for females was 29.8 (SD = 14.1, min = 18, max = 62), and 24.5 for males (SD = 10.4, min = 18, max = 59). Thirty of the participants had completed graduate studies, 7 were pursuing graduate studies, 54 had completed undergraduate studies, 209 were university students, and 8 had only completed high school. All participants gave informed consent, and were treated in accordance with the Code of Ethics of the Psychologist in Mexico [39] and the Declaration of Helsinki [40].

#### 3.1.2. Instruments

In addition to the Brief Scale of Vengeful Tendencies (BSVT) described above and developed in the present study, the following instruments were administered:*TRIM-12, Revenge Motivation Factor* [30]. The validated version for Mexico [56] was used, which has a Cronbach’s alpha of 0.85 for this factor, and 0.895 for the total instrument.*Personal resentment factor of the TFS-CS* [54]. This instrument has an omega report of 0.80. For its application, a translation–retranslation was made by experts, who also verified the cultural adaptation of the instrument.*PPVI, Taking justice into one’s own hands Factor* [55]. This instrument assesses predatory violence, and an omega of 0.772 is reported for the total instrument, and 0.645 for the chosen factor.

#### 3.1.3. Procedure

Participants who responded positively to the invitation were provided with a link to the informed consent form. The questionnaires were arranged on the same internet form, with each questionnaire provided in a different section.

*Statistical analysis*. Relationships between instruments were performed by bivariate correlations using Pearson’s correlation coefficient.

### 3.2. Results

The correlations between each of the scales and the factors and the total BSVT-11 ranged from 0.522 to 0.804, with significant values in all cases (*p* < 0.001). The scale that presented the highest relationships with the BSVT-11 was the TRIM-12. Table 3 illustrates the correlations between the instruments, as well as between each factor. The relationships are significant (*p* < 0.001) in all cases.

### 3.3. Discussion

The concurrent validity analysis shows significantly moderate to strong relationships, thus implying adequate criterion validity. The relationships between the scales allow us to accept the following hypotheses: (1) a positive relationship was found between the revenge planning of the BSVT-11 and the revenge motivation factor of the TRIM-12; (2) a positive relationship was found between the resentment factor of the BSVT-11 and the interpersonal resentment factor of the TFS-CS; (3) the BSVT-11 revenge justification factor was also found to have a positive relationship with the PPVI “Taking justice into one’s own hands” factor; and (4) the total BSVT-11 had positive correlations with each of the concurrent validity scales used. Of note is that the highest correlations of the BSVT and its factors were with the revenge motivation factor of the TRIM-12. This supports the criterion validity of the BSVT-11.

Resentment is the factor of the instrument presented here that shows the highest correlations with the other scales. This makes sense, as resentment is an emotional complex containing anger and hostility, which happens in response to perceived harm, and revenge is the infliction of harm in exchange for the harm suffered [57].

The revenge planning factor correlated highest with the revenge motivation factor from TRIM-12. Though there are similarities between the TRIM-12 and F1 items, the planning factor is not motivation, as in the case of TRIM-12, although it is linked to it. F1 refers to the cognitions that sustain and direct behavior. For it to be motivation, it should include items on the initiation or energization of behavior, since this variable, motivation, is defined as what initiates (or energizes), sustains, and directs behavior [58]. In this study, the factor linked to the initiation or energization of revenge is resentment.

Finally, it is important to note that despite the moderate and significant correlations between the justification factor and the other scales, it is also the factor with the lowest relationships. The scale specifically chosen for the criterion validity of this factor was Factor IV of the PPVI. This instrument has good psychometric properties, but the Likert scale for the response options has four levels, so it has little variability, which may explain the lower correlations.

## 4. General Discussion

The findings of the present study indicate that the BSVT-11 has adequate reliability and validity for assessing revenge tendencies. Revenge is a type of predatory aggression because it is planned or cold-blooded. It is driven by resentment, and requires forms of justification to sustain plans for revenge. Resentment is a motivational force that fosters thoughts of revenge [59] which can take place in diverse settings, ranging from sport hunting [60] to terrorist radicalization [61]. Moreover, because there is an intent to harm, rational or moral justifications are required to make it permissible [62]. On this basis, revenge planning is likely to be more possible. The inventory presented here confirms the presence and relationship between these dimensions of revenge: resentment, justification, and planning, and facilitates their assessment.

Vengeful people are less forgiving and more ruminative [6], which means that people who have suffered some harm present vengeful thoughts both before [16] and after the revenge [8].

It has been pointed out that vengeful people consider revenge to be a moral action [6], and that vengeful acts are appropriate when there is an acceptable justification [5]. In addition, it has been seen that revenge can trigger both positive [8] and negative emotions, for the most part [5]. In this sense, the factors of justification and resentment are supported by the previous points. In addition, it has been suggested that resentment is transformed into a “thirst for revenge” [63], which reaffirms the relationship between the resentment factor and the planning and justification factors. Since revenge has moral elements [6,9,26], justifying it is important, especially since the one who initiates it believes that he/she is doing an act of justice. Revenge represents a direct aggressive behavior towards the individual who committed the injustice [53], which is also considered useful by the one who executes it [8].

The gender differences found are consistent with other studies [52,53]. The gender invariance of the instrument implies that the differences are only related to intensity. However, it has been reported that gender differences in revenge are also due to the type of offense [28]. The instrument does not specify the type of offense, so it is not possible to determine whether the type of harm plays a role. Further studies with this instrument could perhaps explore different populations with different types of harm or offense suffered. Revenge has a mediating role in gender differences in physical aggression [64], and, as such, a possible higher rate of male aggression could not be the explanation for the differences. As in the previously cited study [64], revenge mediates the expression of aggression. Even if personality traits such as agreeableness are incorporated, it has been observed that gender mediates the relationship between this trait and motivation for revenge. Perhaps the development of empathy and forgiveness skills have a differential role by gender. For example, it has been reported that women, compared to men, perceive apologizing and forgiveness as more necessary to reach peace agreements in places where terrorism has been experienced [65]. It is likely that it has been socially reinforced that women are more forgiving, and, therefore, the levels of revenge are lower. In this sense, there is evidence indicating that forgiving women are perceived as more empowered than forgiving men [66].

Although the sample for the temporal stability analysis was small, the results can be used as a guide—both because they reflect the psychometric properties of the instrument, and because they also confirm that revenge is a trait that remains over time [67].

The VSBT-11 was validated in a Mexican sample, but it is likely that the instrument will require few adaptations if used in cultural groups with similar characteristics. Some variables shared by large cultural groups include patterns of individualism–collectivism [68,69], in addition uncertainty avoidance, and power distance [69,70]. Both the first [71] and the last two [72] are reported to be relevant to revenge. Thus, the configuration of the scale may perhaps be similar for Ibero-American cultural groups that share characteristics of the sample studied.

## 5. Limitations

This study only addresses revenge in a general manner. Different types of revenge have been proposed, for example, direct and indirect [73], according to the strategies of revenge, the time course of revenge, or the way revenge is carried out [1]. Further developments of this work could perhaps consider these types of revenge in greater detail. Another important limitation relates to the sample for temporal stability. Though the sample size was small, it can be considered as an indicator. It is recommended that this measure be studied further with a larger sample. This study did not perform mediation or moderation analyses. However, these analyses can be performed in subsequent studies.

It is also important to note that the number imbalance between samples may induce higher fit values in this test. Although they are consistent with the instrument values, the invariance analysis may come from the difference in sample sizes. We have decided to keep the results of the variance analysis as a starting point and comparison for future studies that address this analysis. Another important limitation is that the instrument is validated in a Mexican sample, and, therefore, adaptations are necessary for other populations. As discussed above, this scale may not be generalizable to cultures dissimilar to Mexico. For example, revenge behavior can be direct or indirect depending on the cultural group [74]. It has also been reported that the response to harassment, which may include revenge, is mediated among other variables by the level of individualism–collectivism [75], which are cultural variables. However, given the good psychometric properties of the BSVT-11, it is possible to suggest that future studies can make validations for other samples.

## 6. Conclusions

The main contribution of this study is that this is the first revenge instrument validated in a Mexican sample, which, according to its good psychometric properties and the model used, could be adapted, or used with other populations. For example, its correlations with other instruments related to the construct measured allow us to assume a broad spectrum of application.

Moreover, in a translational sense, due to the reduced number of items, the BSVT-11 allows for a rapid application that could have uses in organizational settings. Healthy work climates require preventive measures. For example, detecting potential retaliatory traits in hired or potential workers may be particularly important in certain work environments. Once a potential for retaliation is known, preventive action can be taken. The shortness of the instrument is an advantage—not only in translational psychology, but also for use in conjunction with other measures without involving participant fatigue (e.g., in the conduct of psychological experiments).

It is important to note that higher rates of resentment and justification of revenge imply greater vengeful thinking. As ruminations increase, the maintenance of revenge increases, which could facilitate the planning of revenge to become a behavior. In this sense, it has been pointed out that once memorized, internal scripts will be more easily retrieved and will guide action in multiple ways [76]. Thus, knowing the indices reflected by the BSVT, and setting forgiveness or understanding of the reasons against it, will likely help to have better psychological interventions on revenge.

Finally, since the prevention of aggression and violence are of the utmost relevance to achieve more peaceful and safe environments, having tools such as the one presented in this study can help achieve such goals of safety and peace.

## Figures and Tables

**Figure 1 behavsci-12-00215-f001:**
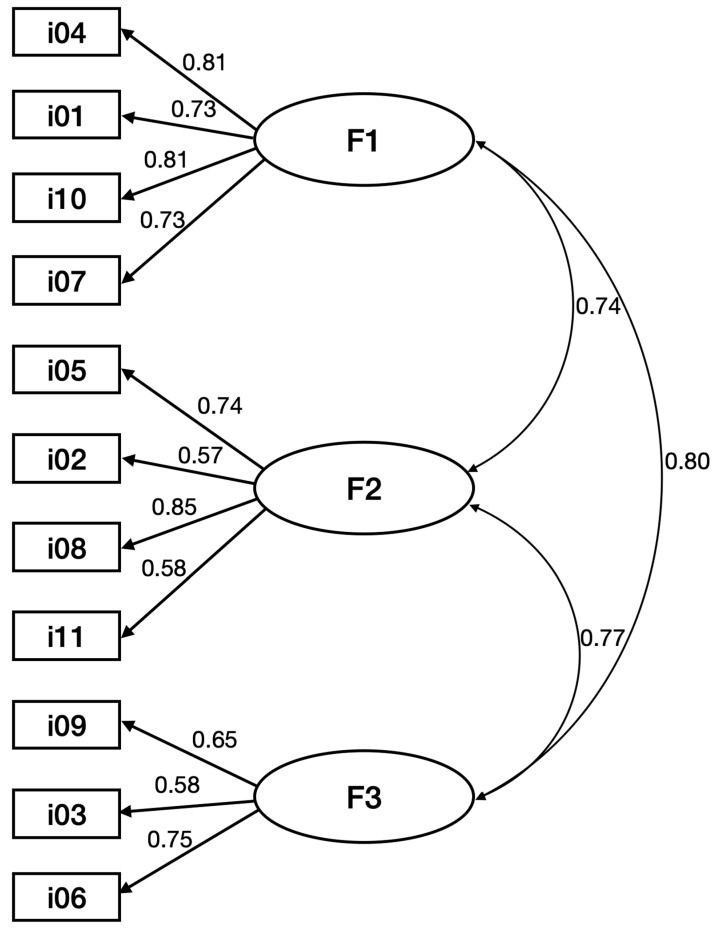
Path diagram with three-factor solution.

**Table 1 behavsci-12-00215-t001:** Factor loadings.

	Factor		
	1	2	3
i04 Planning revenge scenarios gives me satisfaction/Planear escenarios de venganza me produce satisfacción	**0.922**	0.097	−0.15
i07 The thought of hurting the person who hurt me stimulates my imagination/Pensar en lastimar a quien me hizo daño estimula mi imaginación	**0.721**	0.04	0.032
i01 I frequently have thoughts of revenge/Tengo de manera frecuente pensamientos de venganza	**0.692**	0.023	0.066
i10 I often think about harming people who hurt me/Frecuentemente pienso en dañar a las personas que me lastiman	**0.655**	−0.039	0.243
i11 I have thought of revenge as a way of repairing the damage I received/He pensado en vengarme como una forma de reparar el daño que recibí	0.034	**0.864**	−0.065
i02 The damage to my person makes me consider revenge/El daño hacia mi persona me hace considerar la venganza	0.053	**0.593**	0.219
i05 I harbor resentment when someone hurts me/Guardo resentimiento cuando alguien me lastima	−0.078	**0.567**	0.116
i08 I have even committed acts of revenge/He llegado a cometer actos de venganza	0.194	**0.438**	−0.095
i09 I consider it my duty to punish the aggressor if the authority does not do so/Considero que es mi deber castigar al agresor si la autoridad no lo hace	−0.084	0.036	**0.699**
i06 I have felt calm when I hurt my aggressor/Me he sentido tranquilo cuando hago daño a mi agresor	0.253	−0.104	**0.601**
i03 I think revenge is the right thing to do/Pienso que la venganza es un acto correcto	−0.007	0.069	**0.531**

Note: All the factor loadings larger than 0.40 are bold.

**Table 2 behavsci-12-00215-t002:** Descriptive data of the BSVT-11 by gender and total sample.

Gender	F1	F2	F3	BSVT
Women	M = 7.33, SD = 3.24	M = 11.0, SD = 3.77	M = 5.89, SD = 2.31	M = 24.20, SD =7.92
Male	M = 9.06, SD = 3.80	M = 12.5, SD = 3.50	M = 6.89, SD = 2.47	M = 28.4, SD = 8.24
Both	M = 7.85, SD = 3.50	M = 11.4, SD = 3.75	M = 6.19, SD = 2.40	M = 25.5, SD = 8.25

**Table 3 behavsci-12-00215-t003:** Bivariate relationships between the BSVT-11, its factors, and other scales.

	1	2	3	4	5	6
1—F1 Revenge planning	-					
2—F2 Resentment	0.766	-				
3—F3 Justification	0.674	0.717	-			
4— BSVT-11 Total	0.916	0.931	0.853	-		
5—Motivations Factor TRIM-12	0.710	0.762	0.705	0.804	-	
6—Resentment TFS-CS	0.579	0.703	0.522	0.675	0.673	-
7—Taking justice PPVI	0.571	0.556	0.554	0.619	0.544	0.434

## Data Availability

The data presented in this study are available on request from the corresponding author.

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
