# Peer review of "Development and Validation of a Brief Scale of Vengeful Tendencies (BSVT-11) in a Mexican Sample"

_behavsci, 2022, doi:10.3390/bs12070215_

Round 1
Reviewer 1 Report
Dear Authors!
The design of new instruments is always a taugh job and it seems that you did it quite good, but there are several comments and questions that still need to be addressed.
1. The style of the text is at some point really hard, too much of passive constructions that make a paper at some points hardly understandable, particularly in background part. Also you need accurately check for gender neutral descriptions in the text (sometimes speaking of a person you write "he", while one would sound more gender - correct).
2. It would be great to see more detailed description of the sample, particularly - the targeted sample, the age range, why individuals under 18 considered they could participate in your study? Where did they come from, what social and geographic background? In the secind study it looks like (from mean and SD) that participants might have had 14 years old, is that correct?
3. It would be great to know more about the instrument itself, particularly, what was the instruction? What was the scale for estimation of the answers? Likert scale, yes/ no or something else?
4. Could you clarify, why for internal consistensy you used McDonald omega instead of much wider used alpha Cronbach? Is there some rationalle for that?
5. Finally, I suppose (according to authors affiliation) that study was conducted in Mexico, so the original scale is in Spanish. If this is correct, do you think that this scale could be applied in other countries (maybe Iberoamerican), worldwide or maybe they have some contry-specific features that researchers that propapbly will try to adapt to other languages would need to keep in mind? It would be great to have this kind of interpretations in discussion or limitation section.
Author Response
We appreciate your comments. Your valuable observations have enriched our report. Thank you very much.
Point 1 of the review.
Thank you very much for your suggestion to revise the style of the text, in order to improve its comprehensibility and neutrality. We have accepted your suggestion and a native expert has corrected it. The track changes will highlight all these corrections.
Point 2 of the review.
We have improved the wording to clarify that those under 18 years of age were not included in the sample. Line 146.
In addition we added Min and Max years. Line 156
Thank you for your suggestion about more description of sample. We added more information. Line 151-155
Regarding the doubt about the possibility that some participants were 14 years old, due to the standard deviation (Study 2). The minimum age was 18. We have put the minimum and maximum of the sample for clarification. The standard deviation is high because some of our participants were older. Line 424
Point 3 of the review.
Thank you very much for the comment about the lack of mention of how the scale was evaluated and what the instructions were. We have added that it is a Likert scale (Line 169). We have also added the instructions given to the participants (Line 170-225)
Point 4 of the review.
Thank you for the question. El alpha coefficient can be problematic with different loading factors, and is affected by items number (Lines: 272-274)
Point 5 of the review.
We very much welcome your comments and questions on whether the scale can be applied in other countries. Our answer: Yes, with some limitations. The scope is on the 575-581 lines and the limitations are on the 597-604 lines.
Once again, we thank you for your kind and enriching feedback.
Reviewer 2 Report
I have reviewed the paper title: Development and validation of a Brief Scale of Vengeful Tendencies (BSVT-11) in a Mexican sample. The relevant and contributes to existing literature. Therefore, I recommend this paper subject to the following revisions.
- Introduction can be more detailed, including the point of research gap, why researcher must develop BSVT-11 in Maxican sample.
- Methods section is well described and it enables reproducibility. You have put exclusion criteria in the manuscript and it is easy to follow the text.
- Statistic analysis; Do you use License of SPSS and AMOS version allow or not, please reference it.
- Please explain the types of validity and how you were tested in the study.
- EFA and CFA, please explain main point in section, add more detail of your result.
- There is a large difference in the sample size of the two groups. Researchers shall discuss the possibility that the measurement invariance results could be influenced by the difference. Please add more clear.
- The discussion section; should improve your idea discuss and review support more clear point of discussion.
- The manuscript required native enlight edit and proofing, The are a few grammatical and sentence structure errors.
- Please make sure the reference styles, as several references in the bibliography show inconsistent formatting. Some references have the complete name of journals while others have abbreviations. Please update and make it consistent.
This article is strongly recommended for publication after incorporating certain changes. This article needs thorough proofreading. Overall quality of Language is good. Just minor grammatical mistakes are found. All tables and figures are relevant. Research Methodology has been well defined. All data are aligned to the findings of the research. This article is good attempt in the field research and will be beneficial for future researchers.
Author Response
Thank you very much for your careful review and enriching comments. It helped us to improve this study.
On your suggestion about introduction:
Thank you. We have added information about the research gap and therefore the need to create an instrument such as ours. Lines 99-126
In response to your request, we have added the reference for SPSS and Amos (Line 262, reference 45, 46)
About your request on the explanation of the types of validity and how it was tested in the study. We have added a text at the beginning of the procedure section. Lines 231-242.
About more information on EFA y CFA:
We had omitted to report minimum and maximum communalities. We have added them (Lines 308-309). We also added an explanation at the beginning of EFA. (Lines 296-298). In the CFA we have added more values (lines 2322-324) and some text at the beginning of the section (line 316).
We appreciate your deep observation about invariance. We authors have discussed among ourselves and agree with you. We have added your valuable point to the Limitation, in Discussion section. Lines 591-595
Thank you for your suggestion about discussion
We have emphasized the reasoning on the three dimensions proposed and found (lines 514-520) In addition, we have also suggested the scope of use of the inventory (lines 575-581) as well as some limitations (lines 595-604).
Thank you very much for commenting on the need for native editing and proofing. We have taken your suggestion into account and a native expert has corrected the grammatical, stylistic and sentence structure errors.
About reference style:
We also had doubts about the full names of some journals. But they are correct, according to the standard. Journals with only one short word are usually not abbreviated. We use the Behavioral Sciences automatic style tool to enter references via a reference management tool software. Thanks to your suggestion to revise, we have corrected some words that were erroneously capitalized.
Once again, we thank you for your kind and enriching feedback.
Round 2
Reviewer 1 Report
Thank you for your replies, the paper became clearer